Diversity, antibacterial and phytotoxic activities of actinomycetes associated with Periplaneta fuliginosa

Liu Qihua
Tao Jian
Kan Longhui
Zhang Yinglao zhangyl@ahau.edu.cn
Zhang Shuxiang zhangshuxiang90@126.com
School of Life Sciences, Anhui Agricultural University , Hefei , China
Uversky Vladimir
Electronic publication date: 2024 Nov 25
Publication date: 2024
Volume: 12
Electronic Location ID: e18575
Received 2024 Aug 15; Accepted 2024 Nov 1
Copyright: © 2024 Liu et al.
Copyright year: 2024
Copyright holder: Liu et al.
License: This is an open access article distributed under the terms of the Creative Commons Attribution License, which permits unrestricted use, distribution, reproduction and adaptation in any medium and for any purpose provided that it is properly attributed. For attribution, the original author(s), title, publication source (PeerJ) and either DOI or URL of the article must be cited.
License URL: https://creativecommons.org/licenses/by/4.0/

Keywords: Periplaneta fuliginosa, Actinomycetes, Nocardiopsis, Secondary metabolites, Antibacterial activity, Phytotoxic activity

Funding: National Natural Science Foundation of China (NSFC) 32102272 Scientific Research Project of Higher Universities in Anhui Province 2024AH050463 Anhui Agricultural University The School of Life Sciences, Anhui Agricultural University; and Biotechnology center of Anhui Agriculture University This work was supported by the National Natural Science Foundation of China (NSFC) (32102272) and the Scientific Research Project of Higher Universities in Anhui Province (2024AH050463). Qihua Liu received scholarship support from Anhui Agricultural University. The School of Life Sciences, Anhui Agricultural University and the Biotechnology center of Anhui Agriculture University provided convenient laboratory conditions. The funders had no role in study design, data collection and analysis, decision to publish, or preparation of the manuscript.

==============================
Background

Insect-associated actinomycetes represent a potentially rich source for discovering bioactive metabolites. However, the diversity, antibacterial and phytotoxic activities of symbiotic actinomycetes associated with Periplaneta fuliginosa have not yet been conducted.

Results

A total of 86 strains of actinomycetes were isolated from the cornicles and intestines of both nymphs and adults of P. fuliginosa. Diversity analysis revealed that the isolated strains were preliminarily identified as 17 species from two genera, and the dominant genus was Streptomyces. A total of 36 crude extracts (60%) obtained from the supernatant of the 60 fermented strains exhibited a potent antibacterial activity against at least one tested pathogenic bacterium. Among these active strains, 27 crude extracts (75%) exhibited phytotoxic activity against the radicle of Echinochloa crusgalli. Furthermore, seven known compounds, including methoxynicotine (1), (3Z,6Z)-3-(4-methoxybenzylidene)-6-(2-methylpropyl) piperazine-2,5-dione (2), XR334 (3), 1-hydroxy-4-methoxy-2-naphthoic acid (4), nocapyrone A (5), β-daucosterol (6), and β-sitosterol (7) were isolated from an active rare actinomycete Nocardiopsis sp. ZLC-87 which was isolated from the gut of adult P. fuliginosa. Among them, compound 4 exhibited moderate antibacterial activity against Micrococcus tetragenus, Staphylococcus aureus, Escherichia coli, and Pseudomonas syringae pv. actinidiae with the zone of inhibition (ZOI) of 14.5, 12.0, 12.5, and 13.0 mm at a concentration of 30 μg/disc, respectively, which was weaker than those of gentamicin sulfate (ZOI of 29.5, 19.0, 18.5, and 24.5 mm). In addition, the compound 4 had potent phytotoxic activity against the radicle of E. crusgalli and Abutilon theophrasti with the inhibition rate of 65.25% and 92.68% at the concentration of 100 μg/mL.

Conclusion

Based on these findings, this study showed that P. fuliginosa-associated actinomycetes held promise for the development of new antibiotic and herbicide resources.

Introduction

Actinomycetes are widely distributed in ecosystems, thriving in various environments such as soil, marine habitats, and even within organisms, making them an ideal reservoir of natural products (Bao et al., 2021; Hassan & Shaikh, 2017; Protasov et al., 2017). They are capable of synthesizing diverse secondary metabolites including compounds with novel backbones and biological activities (Jose, Maharshi & Jha, 2021). These metabolites possess multiple functionalities such as antimicrobial, antiparasitic, herbicidal, and anticancer activities, holding significant importance in food production, agriculture, and medicine (Olano, Mendez & Salas, 2009; Omura & Crump, 2014; Shi et al., 2020; Wang, Lu & Cao, 2020). However, as exploration of these natural products continues, avenues for discovering them are increasingly depleted. Traditional methods of screening environmental isolates or compound libraries for antibiotic drugs have not yielded new medicines in over three decades (Lewis, 2020). Exploring new sources of natural products has thus become crucial for finding novel compounds. Studies indicate that some insect-associated actinomycetes have genetically diverged from soil and marine actinomycetes by millions of years (McDonald & Currie, 2017), highlighting insect-associated actinomycetes as an underexplored niche with potential for discovering novel bioactive molecules (Van Moll et al., 2021).

Insects represent the most diverse animal group on Earth, estimated at approximately 5.5 million species (Stork, 2018). Microorganisms that live in symbiosis with insects are important source of natural products. Some of these actinomycetes can produce unique types of biologically active compounds (Grundmann et al., 2024). In nature, there are many symbiotic associations between insects and actinomycetes. Numerous studies have shown that a high antimicrobial potential of natural products derived from actinomycetes associated with insects (Chevrette et al., 2019). For instance, southern pine beetles harbor symbiotic actinomycetes that produce polyunsaturated peroxides to mediate the balance of beneficial fungi associated with southern pine beetles (Scott et al., 2008). Metabolic pathways produced by actinomycetes symbiotic with insects differ significantly from those in natural environments, suggesting potential for novel compound discovery. Due to the diversity of insects, they remain an underexploited natural product repository.

Cockroaches are insects that thrive in dark and humid environments, harboring various pathogens (Moges et al., 2016). Their ability to survive under extremely harsh conditions is attributed to the presence of abundant endosymbiotic bacteria within their bodies (Tee & Lee, 2015). 16S rRNA analysis conducted by Vicente, Ozawa & Hasegawa (2016) revealed that a rich microbial community was deposited in the gut of smokybrown cockroach (Periplaneta fuliginosa). Some studies suggest that symbiotic bacteria within cockroaches have the capability to produce antibacterial compounds. For example, Ma et al.’s (2023) work demonstrated that compounds produced by Achromobacter from P. americana could combat multidrug-resistant pathogens such as Klebsiella pneumoniae. The phylum Actinobacteria has a high abundance within cockroaches. Actinobacteria symbiotic with cockroaches have been reported to exhibit various antibacterial activities (Guzman & Vilcinskas, 2020). However, some biological activities of these actinobacteria, such as phytotoxic activities, have rarely been reported. Herbicide resistance has become a significant issue in global agricultural production (Qu et al., 2021). These cockroach-associated actinomycetes with phytotoxic activity may also be a potential source of novel bioherbicides.

In this study, we analyzed the diversity of associated actinomycetes in P. fuliginosa and evaluated their antibacterial and phytotoxic activities. Additionally, the diversity and bioactive properties of metabolites from an active strain Nocardiopsis sp. ZLC-87 were further analyzed.

Materials and Methods

Sample collection and microbial isolations

Twenty healthy nymphs and adults of the P. fuliginosa were randomly grabbed from the garden of Anhui Agricultural University in Hefei, China (longitude: 117°14′57″E, latitude: 31°52′1″N). To prevent contamination, collected samples were stored in sterile centrifuge tubes. Subsequently, samples were promptly transported to the laboratory and subjected to 24 h of starvation treatment. Following starvation, samples were maintained in an environment at −80 °C for the isolation of actinomycetes.

Preparation of samples, the insects were placed into sterile centrifuge tubes containing 30 mL of sterile water for the isolation of the surface actinomycetes (Chevrette et al., 2019). Subsequently, samples were immersed in 30 mL of 75% ethanol for 2 min, followed by three washes with 30 mL of sterile water (30 s each). After external disinfection, samples were dissected using sterile forceps and scalpels to isolate gut tissues. The tissues were then homogenized in 4 mL of sterile water using a mortar and pestle.

Subsequently, the surface washing supernatant and the homogenate were serially diluted in 10-fold increments (10−1, 10−2, 10−3), and 100 µL of each sample was spread onto six different culture media, including ISP 2 agar (ISP2), chitin agar (CA), starch-casein agar (SCA), GYM Streptomyces agar (GYM), Gause’s No. 1 agar (GS), and modified HV agar (M-HV). All media were supplemented with nalidixic acid (50 mg/L) and nystatin (25 mg/L) to inhibit the growth of Gram-negative bacteria and fungi (Li et al., 2021).

The cultures were then incubated at 28 °C for 1–4 weeks. Actinomycete colonies were subsequently transferred to solid Gause’s No. 1 agar (GS), ISP 2 agar (ISP2) or GYM agar, and then preserved in slants form at 4 °C or as 25% (v/v) glycerol suspensions stored at −80 °C until further use.

Molecular characterization of the isolates

The isolated strains were initially identified based on their morphological characteristics. For molecular identification, genomic DNA of actinomycetes was amplified using universal primers 27F (5′-TCCTCCGCTTATTGATATGC-3′) and 1492R (5′-GGTTACCTTGTTACGACTT-3′) the 16S rRNA gene (Long et al., 2022). All PCR products were validated by 2% agarose gel electrophoresis. Subsequently, PCR products were sent to Tsingke Biotechnology Co., Ltd. (Beijing, China) for sequencing. The obtained sequences were compared with closely related reference strains to achieve the highest sequence similarity (species description) with type strain sequences, by the EzTaxon-e server (https://www.ezbiocloud.net/) (Kim et al., 2012).

Based on the 16S rRNA sequence alignment and evolutionary tree analysis (Fig. S2), the selected strain ZLC-87 for further study was named as Nocardiopsis sp. ZLC-87.

The obtained gene sequences were deposited in the GenBank database with accession numbers PP456280–PP456365.

Preparation the extracts of fermentation broth of associated actinomycetes

Based on morphological characteristics and molecular identification, 60 selected strains were screened for small-scale fermentation to isolate actinomycetes with antibacterial activity. Among them, Streptomyces was cultured in GS liquid media, while Nocardiopsis was cultured in liquid GYM medium. Cultures were incubated at 28 °C with agitation at 180 rpm for 7 days. After cultivation, the cultures were filtered to obtain the broth (300 mL), followed by liquid-liquid extraction with an equal volume of ethyl acetate (EtOAc), repeated three times. The organic phase extracts were collected, concentrated under vacuum, solvent evaporated, resulting in crude extracts of the fermented broth.

Isolation of secondary metabolites

Following small-scale fermentation and activity screening, strains demonstrating potent antibacterial activity and abundant secondary metabolite production were selected for isolation and identification of their products. In this study, the active strain ZLC-87 was chosen for compound purification and characterization. The strain was inoculated into GYM media and cultivated at 28 °C for 7 days with agitation at 180 rpm to prepare seed cultures. Subsequently, seed cultures were transferred to solid rice medium composed of rice and deionized water in a ratio of 1.0:1.1 (w/w), sterilized under high pressure at 120 °C for 30 min. Fermentation was carried out in 50 Erlenmeyer flasks, each containing 100 g of rice and 110 mL of water, incubated at 28 °C for approximately 2 months.

Following fermentation, the rice was soaked in methanol and subjected to ultrasonic extraction for 20 min, followed by a 24-h soaking period. After removing a significant portion of methanol by rotary evaporation and adding water (1 L), extraction was performed using an equal volume of ethyl acetate (EtOAc). The resulting organic phase was evaporated under vacuum rotary evaporation to yield crude extracts of the product (3.86 g). The crude extract was subjected to silica gel (100–200 mesh) column chromatography, eluting with a gradient of CH2Cl2/MeOH (100:0–100:32, v/v) to obtain six fractions (Fr1–Fr6). Fractions were monitored by thin-layer chromatography (TLC), and further purification was achieved using Sephadex LH-20 column chromatography and high-performance liquid chromatography (HPLC) to obtain purified compounds. Fr2 (CH2Cl2/MeOH, 100:2, v/v) was further purified on a silica gel column and eluted with a CH2Cl2/MeOH mixture (100:1, v/v) to yield compound 1 (18 mg) and compound 3 (16 mg). Compound 2 (25 mg) precipitated from the CH2Cl2 solution of Fr3 (CH2Cl2/MeOH, 100:4, v/v). Fr4 (CH2Cl2/MeOH, 100:8, v/v) was further purified on a silica gel column and eluted with a CH2Cl2/MeOH mixture (100:2, 100:8, v/v) to obtain compound 4 (10 mg) and compound 5 (9 mg). Fr5 (CH2Cl2/MeOH, 100:16, v/v) was separated using a Sephadex LH-20 column with methanol as the mobile phase to yield crystals of compound 6 (17 mg). Compound 7 (13 mg) crystallized from the petroleum ether solution of Fr2 (CH2Cl2/MeOH, 100:1, v/v).

Structural elucidation of metabolites

All compounds were structurally analyzed preliminarily using 1H/13C nuclear magnetic resonance (NMR) spectroscopy and high-resolution electrospray ionization mass spectrometry (HR-ESI-MS). 1H/13C NMR data were collected on an Agilent DD2 600 Hz spectrometer (Agilent, Santa Clara, CA, USA), with chemical shifts referenced to tetramethylsilane (TMS) and reported in parts per million (δ). HR-ESI-MS spectra were acquired using a mass spectrometer in TripeTOF 4600 mass system (Bruker, Billerica, MA, USA).

Antimicrobial activities

The filter paper disc method was employed to evaluate the antibacterial activity of crude extracts (Balouiri, Sadiki & Ibnsouda, 2016). Test pathogens included Micrococcus tetragenus ATCC35098, Staphylococcus aureus ATCC6538, Escherichia coli ATCC8739, and Pseudomonas aeruginosa syringae pv. actinidiae (Psa). The crude extracts dissolved in acetone and filtered through a sterile 0.22 μm filter membrane (at a concentration of 30 μg/disc) was added to agar plates and incubated at 37 °C. Each crude extracts were tested in triplicate, with acetone and gentamicin sulfate used as negative and positive controls, respectively. The method for the compounds was the same as described above. When compounds were dissolved using non-acetone solvents, the corresponding solvent was used as the negative control. Antimicrobial activity was assessed by measuring the zone of inhibition (ZOI).

Phytotoxic activity

After screening for antibacterial testing, 36 actinomycetes with antibacterial activity were selected for phytotoxic activity testing. According to the methods described in previously (Sun et al., 2020; Zhang et al., 2011), the phytotoxic activity of actinomycetes was evaluated on radicle growth of Echinochloa crusgalli. Crude extracts were dissolved in DMSO and diluted to a concentration of 100 μg/mL with 0.1% Tween 80 aqueous solution. Seeds of E. crusgalli were immersed in 5% sodium hypochlorite solution for 20 min for surface sterilization. After several rinses with deionized water, seeds were transferred to a 28 °C illuminating incubator until germination. Subsequently, 20 pre-germinated seeds were placed in petri dishes (90 mm diameter) containing filter paper discs, and 5 mL of the compound solution was added. The length of seedling roots was measured after 2–3 days. Distilled water served as a negative control. 2,4-dichlorophenoxyacetic acid (2,4-D) was used as positive control.

All compounds were evaluated for phytotoxic activity against E. crusgalli and Abutilon theophrasti using Petri dish bioassay. Seeds of E. crusgalli were germinated using the method described above. Seeds of A. theophrasti were immersed in water at 60 °C for 30 min followed by soaking in 40 mmol/L CaCl2 solution for 12 h, and then subjected to similar surface sterilization methods as described above. Then, seeds were transferred to a 28 °C illuminating incubator until germination. The compounds were dissolved as described above. The bioassay of the phytotoxic activity for compounds was the same as that of crude extracts but using 30 seeds. 2,4-D was used as a positive control.

Result

Isolation and identification of associated actinomycetes

This study isolated 86 strains of actinomycetes from the P. fuliginosa using six different culture media (Fig. S1, Table S2). Among these, nine strains were isolated from larval cuticle, 29 from larval gut, 12 from adult cuticle, and 36 from adult gut. The majority of isolates were obtained from HV medium (23 strains, 26.7%) and SCA medium (19 strains, 22.1%), followed by CA (17 strains, 19.8%), GS (13 strains, 15.1%), ISP2 (eight strains, 9.3%), and GYM (six strains, 7.0%). Therefore, HV and SCA media were conducive to the isolation of actinomycetes.

16S rRNA sequencing was employed to identify all isolated strains, resulting in a total of 17 species from two genera. Among them, 85 strains belonged to the dominant genus Streptomyces (Fig. 1), with an isolation frequency of 98.84%, while the remaining one strain belonged to the genus of rare Nocardiopsis, with an isolation frequency of 1.16%.

Figure 1 Frequency of isolation of associated actinomycetes of P. fuliginosa.

Differential analysis of associated actinomycetes

Based on the 16S rRNA sequences alignment results from the data of EzBioCloud, each isolated strain was classified as the closest similarity type strain species, separately. The species of actinomycetes were preliminary counted (Fig. 2). The results showed that the actinomycetes isolated from the gut exhibited higher diversity compared to those isolated from the cuticle. Among the 65 gut-derived strains, a total of 16 species were identified. Apart from the dominant strain Streptomyces pratensis, which accounted for 30% of gut isolates, the distribution of other strains was relatively even. In contrast, among the 21 strains isolated from the cuticle, only four species were identified, with Streptomyces cavourensis (43%) and S. pratensis (48%) being overwhelmingly dominant. This indicates lower diversity among cuticle-associated actinomycetes, highlighting greater diversity of symbiotic actinomycetes within the gut.

Figure 2 Venn diagram analysis of isolated culturable actinomycetes from P. fuliginosa.

Adult and larval samples also exhibited significant differences in actinomycete isolation. A total of 48 strains were isolated from adult samples, comprising 12 species, whereas 38 strains from larval samples included 10 species. There were overlaps between actinomycetes isolated from adults and nymphs, indicating both shared and distinct populations.

Antibacterial activity of the crude extracts of the isolated symbiont

The antibacterial activity of crude extracts obtained from 60 isolated strains of actinomycetes was evaluated using the filter paper disc method (Fig. 3). The results indicated that 36 extracts of fermented strains exhibited inhibitory activity against one or more pathogens. Specifically, 35 strains showed inhibition against M. tetragenus and among these effective strains, ZLC-65, ZLC-81, and ZLC-85 exhibited significant inhibitory effects against M. tetragenus, with ZOI of 26.5, 27.0, and 25.5 mm, which were slightly weaker than that of the gentamicin sulfate (ZOI of 27.5 mm). A total of 29 strains showed inhibition against S. aureus, and four of these strains, including ZLB-32, ZLC-97, ZLC-101, and ZLC-105 showed notable inhibitory activity, with ZOI values exceeding 14 mm, albeit lower than that of the positive control gentamicin sulfate (ZOI of 19.0 mm). A total of 26 strains showed activity against Psa, with most strains exhibiting moderate inhibition. Except for ZLC-65, none of the strains showed inhibitory activity against E. coli. Notably, extracts from strain ZLC-65 exhibited inhibitory activity against all four test pathogens (M. tetragenus, S. aureus, E. coli, and Psa), with ZOI of 26.5, 12.0, 13.0, and 10.5 mm, respectively. Further analysis indicated that extracts from actinomycetes showed significantly higher inhibitory activity against Gram-positive bacteria compared to Gram-negative bacteria.

Figure 3 Heatmap of ZOI for antimicrobial activity of 60 strains.

a, Gentamicin sulfate as the positive control of pathogenic bacteria; the concentration for the test is 30 μg/filter paper.

Phytotoxic assay

A total of 36 actinomycetes with antibacterial activity were selected for phytotoxic activity testing. As the results shown in Table 1, 27 crude extracts of actinomycetes exhibited phytotoxic activity against the radicle of E. crusgalli, at the concertation of 100 μg/mL. Among these strains, one strain showed strong phytotoxic activity against E. crusgalli with the inhibition rate of 100%. Eight strains showed good phytotoxic activity against E. crusgalli with the inhibition rate of 80–99%. A total of 13 strains showed potent phytotoxic activity against E. crusgalli with the inhibition rate of 60–79%. In addition, five strains showed relatively weak phytotoxic activity against E. crusgalli with the inhibition rate less than 60%.

Table 1 Phytotoxic effects of the crude extracts of the selected actinomycetes on the radicle growth of E. crusgalli.

Strains	Inhibition rate/%	Strains	Inhibition rate/%	
ZLB-11	87.02 ± 0.53	ZLC-52	57.09 ± 1.23	
ZLB-12	71.80 ± 1.55	ZLC-56	76.64 ± 0.96	
ZLB-14	64.53 ± 1.04	ZLC-57	55.88 ± 1.19	
ZLB-15	85.29 ± 0.63	ZLC-58	66.26 ± 0.94	
ZLB-16	51.90 ± 1.29	ZLC-59	81.83 ± 0.76	
ZLB-18	91.70 ± 0.72	ZLC-65	67.47 ± 1.18	
ZLB-21	NIa	ZLC-81	NI	
ZLB-23	64.53 ± 1.05	ZLC-85	77.85 ± 1.01	
ZLB-24	100.00 ± 0.00	ZLC-87	89.62 ± 0.72	
ZLB-27	81.83 ± 0.67	ZLC-95	NI	
ZLB-32	NI	ZLC-96	NI	
ZLC-45	59.69 ± 1.08	ZLC-97	NI	
ZLC-46	87.02 ± 0.72	ZLC-101	NI	
ZLC-47	78.37 ± 0.95	ZLC-102	NI	
ZLC-48	83.56 ± 0.76	ZLC-103	NI	
ZLC-49	79.58 ± 0.65	ZLC-105	74.39 ± 1.09	
ZLC-50	75.26 ± 0.91	ZLC-106	58.82 ± 1.20	
ZLC-51	65.74 ± 0.64	ZLC-107	65.40 ± 0.83	
Notes:

Results are presented as inhibition rate % (mean ± standard error).

a “NI” means not inhibited; the concentration for the test is 100 μg/mL.

Identification of secondary metabolites isolated from ZLC-87

Seven known compounds (Fig. 4) were isolated and purified from the fermentation products of selected active strain ZLC-87. Compounds 1–7 were identified as methoxyneihumicin (1) (Zhang et al., 2013), (3Z,6Z)-3-(4-Methoxybenzylidene)-6-(2-methylpropylidene) piperazine 2,5-dione (2) (Sun et al., 2017), XR334 (3) (Bryans et al., 1996), 1-Hydroxy-4-methoxy-2-naphthoic acid (4) (Pfefferle et al., 1997), nocapyrone A (5) (Schneemann et al., 2010), β-daucosterol (6) (Luo et al., 2009) and β-sitosterol (7) (Kamal et al., 2016) by comparing the NMR and MS data with those reported in the literatures.

Figure 4 The chemical structures of compounds 1–7.

Compound (1): white crystal; HR-ESI-MS: m/z 334.1314 [M+H]+, calculated for C20H18N2O3 334.1317; 1H NMR (600 MHz, CDCl3) δ 8.11 (d, J = 8.3 Hz, 2H), 8.03 (s, 1H), 7.43 (d, J = 7.6 Hz, 2H), 7.37 (d, J = 7.5 Hz, 2H), 7.33 (t, J = 14.6 Hz, 2H), 7.28 (s, 1H), 6.94 (d, J = 8.2 Hz, 2H), 6.56 (s, 1H), 4.06 (s, 3H), 3.86 (s, 3H). 13C NMR (150 MHz, CDCl3) δ 54.6, 55.5, 110.2, 114.1, 124.7, 128.2, 128.5, 128.5, 128.7, 129.6, 133.6, 133.7, 153.8, 160.5, 160.6.

Compound (2): white powder, HR-ESI–MS: m/z 286.1311 [M+H]+, calculated for C16H18N2O3 286.1317; 1H NMR (600 MHz, DMSO-d6) δ 10.27 (s, 1H), 9.89 (s, 1H), 7.47 (d, J = 6.2 Hz, 2H), 6.97 (d, J = 6.3 Hz, 2H), 6.70 (s, 1H), 5.68 (d, J = 10.4 Hz, 1H), 3.78 (s, 3H), 2.95 (s, 1H), 0.98 (d, J = 3.7 Hz, 6H). 13C NMR (150 MHz, DMSO-d6) δ 159.6, 158.3, 158.0, 131.3, 126.0, 125.8, 125.7, 125.3, 114.9, 114.7, 55.7, 40.6, 40.0, 24.4, 22.7.

Compound (3): green solid, HR-ESI–MS: m/z 322.1303 [M−H]−, calculated for C19H18N2O3 322.1317; 1H NMR (600 MHz, CDCl3) δ 8.13 (s, 1H), 7.46 (d, J = 7.3 Hz, 2H), 7.38 (d, J = 12.3 Hz, 2H), 7.36 (s, 2H), 7.24 (s, 3H), 7.04–6.95 (m, 3H), 3.85 (d, J = 11.0 Hz, 2H). 13C NMR (150 MHz, CDCl3) δ 157.37, 157.19, 132.97, 130.21, 129.73, 129.10, 128.58, 125.22, 124.46, 117.03, 116.47, 115.26.

Compound (4): yellow crystal, HR-ESI-MS: m/z 218.0581 [M+H]+, calculated for C12H10O4 218.0579; 1H NMR (600 MHz Acetone-d6) δ 8.34 (d, J = 8.3 Hz, 1H), 8.19 (d, J = 8.4 Hz, 1H), 7.68 (t, J = 7.6 Hz, 1H), 7.61 (t, J = 7.4 Hz, 1H), 7.19 (s, 1H), 3.99 (s, 3H). 13C NMR (150 MHz, Acetone-d6) δ 173.73, 156.50, 148.63, 130.86, 129.85, 127.39, 126.55, 124.46, 122.91, 106.10, 102.61, 56.32.

Compound (5): colorless solids, HR-ESI–MS: m/z 268.1670 [M+H]+, calculated for C15H24O4 268.1675; 1H NMR (600 MHz, CDCl3) δ 3.95 (s, 3H), 2.60 (t, J = 7.6 Hz, 2H), 1.94 (s, 3H), 1.84 (s, 3H), 1.68–1.63 (m, 3H), 1.51–1.48 (m, 2H), 1.45 (dq, J = 7.0, 4.2, 3.8 Hz, 2H), 1.30 (s, 6H), 1.25 (s, 6H), 1.22 (s, 6H), 0.91–0.82 (m, 3H). 13C NMR (150 MHz, CDCl3) δ 181.19, 118.48, 99.69, 70.91, 55.46, 43.66, 30.93, 29.84, 29.55, 27.76, 24.08, 10.08, 6.99.

Compound (6): colorless crystal, HR-ESI–MS: m/z 576.4375 [M+Na]+, calculated for C35H60O6 576.4390; 1H NMR (600 MHz, DMSO-d6) δ 5.32 (s, 1H), 4.91 (d, J = 10.9 Hz, 2H), 4.86 (s, 1H), 4.44 (s, 1H), 4.22 (d, J = 7.6 Hz, 1H), 3.63 (s, 1H), 3.50–3.38 (m, 2H), 3.13 (s, 1H), 3.05 (s, 2H), 2.90 (s, 1H), 2.36 (d, J = 12.2 Hz, 1H), 2.16–2.08 (m, 1H), 1.99–1.88 (m, 2H), 1.80 (s, 3H), 1.63 (d, J = 6.3 Hz, 1H), 1.53–1.46 (m, 4H), 1.40 (d, J = 10.7 Hz, 2H), 1.35–1.28 (m, 2H), 1.26–1.20 (m, 2H), 1.18–1.12 (m, 3H), 0.98 (s, 3H), 0.95 (s, 3H), 0.92–0.88 (m, 4H), 0.84–0.78 (m, 8H), 0.65 (s, 3H). 13C NMR (150 MHz, DMSO-d6) δ 140.44, 121.09, 100.77, 76.92, 76.75, 76.67, 73.43, 70.09, 61.05, 56.13, 55.41, 49.58, 45.13, 41.81, 40.06, 38.29, 36.78, 36.16, 35.40, 33.33, 31.38, 31.31, 29.22, 28.71, 27.70, 25.49, 23.79, 22.58, 20.54, 19.63, 19.02, 18.90, 18.56, 11.72, 11.60.

Compound (7): colorless crystal, HR-ESI–MS: m/z 414.2042 [M+Na]+, calculated for C29H50O 414.3862; 1H NMR (600 MHz, Acetone-d6) δ 5.31 (s, 1H), 3.39 (s, 1H), 2.24–2.17 (m, 1H), 2.00–1.92 (m, 1H), 1.87–1.80 (m, 1H), 1.80 1.73 (m, 1H), 1.73–1.64 (m, 1H), 1.64–1.58 (m, 1H), 1.58–1.53 (m, 2H), 1.53–1.49 (m, 1H), 1.50–1.43 (m, 2H), 1.43–1.38 (m, 1H), 1.35–1.27 (m, 2H), 1.25–1.18 (m, 2H), 1.09–1.03 (m, 3H), 0.97 (d, J = 6.4 Hz, 3H), 0.88–0.83 (m, 5H), 0.83 (s, 2H), 0.76–0.71 (m, 2H).13C NMR (150 MHz, Acetone-d6) δ 142.44, 121.57, 71.69, 57.74, 57.02, 51.31, 46.82, 43.34, 40.74, 38.30, 37.38, 36.97, 34.79, 32.87, 32.68, 32.53, 30.34, 30.03, 28.98, 26.92, 24.99, 23.84, 21.85, 20.11, 19.83, 19.40, 19.28, 12.32, 12.28.

Antibacterial activities of compounds

The inhibitory activity of isolated compounds against four pathogens listed in Table 2. The results indicate that compound 4 exhibited moderate inhibitory activity against the pathogens M. tetragenus, S. aureus, E. coli, and Psa, with ZOI of 14.5, 12.0, 12.5, and 13.0 mm, respectively, which were weaker than that of positive control (ZOI of 29.5, 19.0, 18.5, and 24.5 mm, respectively). Compounds 1, 2, 3, 5, 6, and 7 showed no antimicrobial activity.

Table 2 Inhibitory effects of compounds 1–7 isolated from ZLC-87 against the tested bacteria.

Compound	M. tetragenus	S. aureus	E. coli	P. syringae pv. actinidiae	
1	NIb	NI	NI	NI	
2	NI	NI	NI	NI	
3	NI	NI	NI	NI	
4	14.5 ± 0.38	12.0 ± 0.58	12.5 ± 0.58	13.0 ± 0.50	
5	NI	NI	NI	NI	
6	NI	NI	NI	NI	
7	NI	NI	NI	NI	
Gentamicin sulfatea	29.5 ± 0.14	19.0 ± 0.52	18.5 ± 0.29	24.5 ± 1.06	
Notes:

Results are presented as the ZOI (mm, mean ± standard error).

a Gentamicin sulfate as the positive control of pathogenic bacteria.

b “NI” means not inhibited; the concentration for the test is 30 μg/filter paper.

Phytotoxic activities of compounds

The phytotoxic activity of secondary metabolites from Nocardiopsis was evaluated against E. crusgalli and A. theophrasti using Petri dish bioassay (Table 3). Results demonstrated that compound 4 exhibited inhibitory effects against A. theophrasti (Fig. S3), with inhibition rates of 92.68% at a concentration of 100 μg/mL, which was weak than that of 2,4-D with an inhibition rate of 100%. Compound 4 also showed a moderate inhibition against E. crusgalli with inhibition rate of 65.25%. In addition, compound 1 exhibited moderate inhibition against A. theophrasti, with inhibition rate of 73.13%. Furthermore, compounds 1, 2, 5, 6 and 7 exhibited weak inhibition against E. crusgalli, with inhibition rates less than 47%.

Table 3 Phytotoxic activities of compounds 1–7 on the radicle growth of E. crusgalli and A. theophrasti.

Compound	E. crusgalli	A. theophrasti	
1	25.31 ± 4.78	73.13 ± 1.60	
2	24.69 ± 3.96	NI	
3	NIb	NI	
4	65.25 ± 4.12	92.68 ± 2.44	
5	45.74 ± 7.90	NI	
6	46.44 ± 4.73	NI	
7	11.38 ± 6.00	10.33 ± 5.09	
2, 4-Da	100.00 ± 0.00	100.00 ± 0.00	
Notes:

Results are presented as the inhibition rate % (mean ± standard error).

a 2,4-D as the positive control of phytotoxic activity.

b “NI” means not inhibited; the concentration for the test is 100 μg/mL.

Discussion

In this study, the diversity of cuticle and gut associated actinomycetes of P. fuliginosa which in different growth stages was studied for the first time. A total of 86 culturable actinomycetes distributed in 17 species from two genera were isolated and preliminarily identified. Similar to culture-independent community analysis in the literature (Vicente, Ozawa & Hasegawa, 2016), P. fuliginosa possessed a high diversity of gut microbiome. Differential analysis revealed higher diversity in adult samples compared to larval samples and higher diversity in gut compared to that of cuticle. Factors such as the interaction period with the diets or habitat may lead to these differences (Lee et al., 2020).

The dramatic increase in antibiotic-resistant bacteria has challenged global healthcare systems like never before. Symbiotic actinomycetes from arthropods, represented by insects, are an under-exploited natural product reservoir that could help break the antibiotic resistance crisis (Olano & Rodriguez, 2024). As an insect that survives in harsh environments, cockroach-associated actinomycetes play a critical role in the robust growth of cockroach in highly microbially polluted environments which have the potential to develop novel antibiotics (Kaltenpoth, Engl & Clay, 2013). In this study, 60% of strains evaluated for antimicrobial activity exhibited antibacterial effects. Similar to other species of cockroaches (Akbar et al., 2018), P. fuliginosa symbiotic microorganisms, especially those in the gut, have an inhibitory effect on pathogenic bacteria. In addition, P. fuliginosa-associated actinomycetes were more effective in inhibiting Gram-positive bacteria compared to other cockroach symbiotic bacteria. Therefore, P. fuliginosa-associated actinomycetes might be a potential antimicrobial resource.

Biological herbicides, due to their environmental friendliness and sustainability, have emerged as a promising alternative to conventional chemical herbicides. Research has revealed that certain actinomycetes strains can effectively inhibit the growth of E. crusgalli, a major weed in rice fields, without causing harm to the cultivated plants (Lee et al., 2003). We investigated the phytotoxic activity of crude extracts from all strains with antibacterial activity, and 75% of the crude extracts showed inhibitory effects on E. crusgalli. These findings suggest that P. fuliginosa-associated actinomycetes could play a pivotal role in the development of eco-friendly herbicides, and understanding these aspects may pave the way for the widespread application of actinomycetes-based herbicides.

Furthermore, we investigated the secondary metabolites from one rare actinomycete-Nocardiopsis strain ZLC-87 with outstanding antibacterial and phytotoxic activities. Nocardiopsis possesses the capability to produce novel natural products, with some studies indicating its wide applications in medicine, agriculture, and environmental protection (AbdElgawad et al., 2021; Bennur et al., 2016; Khalil et al., 2021; Patel et al., 2021). In this study, seven secondary metabolites were obtained from ZLC-87. Among them, compounds 1, 2, and 3 are diketopiperazines (DKPs), a structural class of compounds previously reported to exhibit a variety of biological activities (Liu et al., 2024). However, the three DKPs isolated from ZLC-87 did not exhibit antibacterial activity, which is consistent with the findings reported by Zhang et al. (2013). Additionally, compounds 1 and 2 demonstrated moderate phytotoxic activity. Further investigation into the DKPs produced by Nocardiopsis is warranted. Compound 4 demonstrated inhibitory activity against all four tested pathogens, similar to the antibacterial activity reported for the same compound isolated from N. aegyptia by Yongjun et al. (2023). Furthermore, studies have shown that compound 4 also possesses phytotoxic activity against Lemna minor (Pfefferle et al., 1997). Here, we evaluated the phytotoxic activity of compound 4 against two common agricultural weeds, E. crusgalli and A. theophrasti. The results indicate that compound 4 exhibits significant inhibitory activity against both weeds. The phytotoxic activity and the antimicrobial activity against Psa of compound 4, suggest potential for further research into its antimicrobial and herbicidal properties as a basis for developing novel agricultural antibiotics.

Conclusion

In this study, we analyzed the diversity of P. fuliginosa associated actinomycetes, and revealed that cockroaches are potential sources for screening culturable bioactive actinomycetes. Bioactive assays showed the antimicrobial and phytotoxic activity by many of these cockroach-associated actinomycetes. Furthermore, we isolated seven known compounds from Nocardiopsis sp. ZLC-87 among which one compound demonstrated antibacterial activity against four pathogenic bacteria, while six compounds exhibited varying degrees of phytotoxic activity on two weeds. These studies indicate that P. fuliginosa-associated actinomycetes are a valuable source of bioactive secondary metabolites, warranting further exploration of their potential applications in biocontrol.

Supplemental Information

Supplemental Information 1 Supplementary Figures and Tables.

Supplemental Information 2 Raw data: phytotoxic activity.

The medium, phylogenetic analysis of all the isolates, colony morphology of some of the actinomycetes, phytotoxic activity effect graph of compound 4, NMR and MS data of all the compounds.

Additional Information and Declarations

Competing Interests

Author Contributions

DNA Deposition

Data Availability

The authors declare that they have no competing interests.

Qihua Liu performed the experiments, analyzed the data, prepared figures and/or tables, authored or reviewed drafts of the article, and approved the final draft.

Jian Tao analyzed the data, prepared figures and/or tables, authored or reviewed drafts of the article, and approved the final draft.

Longhui Kan analyzed the data, authored or reviewed drafts of the article, and approved the final draft.

Yinglao Zhang conceived and designed the experiments, authored or reviewed drafts of the article, and approved the final draft.

Shuxiang Zhang conceived and designed the experiments, authored or reviewed drafts of the article, and approved the final draft.

The following information was supplied regarding the deposition of DNA sequences:

The obtained gene sequences are available at GenBank: PP456280–PP456365.

The following information was supplied regarding data availability:

The raw data are available in the Supplemental File.

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
