# Peer review of "Diversity, antibacterial and phytotoxic activities of actinomycetes associated with Periplaneta fuliginosa"

_PeerJ, doi:10.7717/peerj.18575_

## Round 0.1 · original submission · Major Revisions

Please address issues pointed by all reviewers and amend manuscript accordingly.

Reviewer 2 ·

Basic reporting

This manuscript is well-written and provides comprehensive data on cockroach-derived actinomycetes. After reviewing the manuscript, I have a few suggestions that could further enhance the quality of the work, as outlined below.

Experimental design

- as provided in the additional comments

Validity of the findings

- as provided in the additional comments

Additional comments

Introduction: The rational for screening the phytotoxic activity of the cockroach-derived actinomycetes should be added into the introduction.
Line 80: The method for collecting the samples and number of cockroaches used in this study should be added to this line.
Line 95: Streptomyce Agar --> Streptomyces [italic] agar
Line 109-110 model strain --> type strain
Line 150-151:Use the full name for genus because this is the first mentioned in the main text and remove the bracket between the strain name. “…….Micrococcus tetragenus ATCC 35097, Staphylococcus aureus ATCC 6538, Escherichia coli ATCC 8739, Pseudomonas aeruginosa syringae pv. actinidiae (Psa).” In addition, added the strain number for Psa.
Line 152: I am uncertain whether acetone is able to fully dissolve the crude extract. If the authors employed any methods to remove undissolved material, please provide additional details on the procedure.
Line 157: I question the rationale behind selecting only the strain that exhibited antimicrobial activity for further phytotoxic activity testing. Is there any evidence suggesting a relationship between phytotoxic and antimicrobial activities? If no such relationship exists, I recommend that all strains should be tested for phytotoxic activity to ensure a comprehensive evaluation.
Line 160: E. crusgalli --> Echinochloa crusgalli [italic]
Line 168: A. theophrasti --> Abutilon theophrasti [italic]
Line: 175: 2,4-dichlorophenoxyacetic acid (2,4-D) --> 2,4-D
Line 192, 194: “S.” --> “Streptomyces” because this is the first mentioned of this species.
Line 280-284: In my opinion, authors should determine the MIC of the pure compound (4) instead of reporting only the ZOI.
Figure 4: Please redraw the ring D of the steroid structures in compounds (6) and (7) to ensure consistency in the presentation.

Reviewer 3 ·

Basic reporting

Thanks for giving me the opportunity to review the manuscript (#104661) titled "Diversity, antibacterial and phytotoxic activities of actinomycetes associated with Periplaneta fuliginosa" for consideration the the journal PeerJ. However, it is import for the authors to consider the following issues for the impovement of the manuscript:
Line 67: provide full common names and scientific name with authority. That is smokybrown cockroach (Periplaneta fuliginosa (Serville, 1839)).

Figure 4: Please name these compounds in brackefs in each number. That is methoxynicotine (1), (3Z,6Z)-3-(4-methoxybenzylidene)-6-(2-methylpropyl) piperazine-2,5-
dione (2), XR334 (3), 1-hydroxy-4-methoxy-2-naphthoic acid (4), nocapyrone A (5), ³-daucosterol (6),
and ³-sitosterol (7).

Table 1 caption: This should be "standard error" not "standard". Also include mean ± standard error in brackets. That inhibition rate % (mean ± standard error).

Table 2 caption : This should be "standard error" not "standard". Also include mean ± standard error in brackets. That inhibition rate % (mean ± standard error).

Table 3 caption : This should be "standard error" not "standard". Also include mean ± standard error in brackets. That inhibition rate % (mean ± standard error).

Experimental design

The materials and methods presented in this manuscript are clear, and in line with the aim.

Validity of the findings

The findings are valid. The captions of tables 1-3 should be improved

Reviewer 4 ·

Basic reporting

This manuscript distribute diversity of insect-associated actinobacteria from Periplaneta fuliginosa, and present the known metabolites from the selected strain. The authors found that Streptomyces were major genus but only one genus Nocardiopsis was a rare actinomycetes. In the results, the selected strain ZLC-87 was identified to Nocardiopsis alba that based on morphological and 16S rRNA gene analysis could not confirm to the species level, this result should be show only genus level and similarity value of closely related species.

Experimental design

Lines 86-91; “According to Chevrette et al. (Chevrette et al. 2019), samples were placed into sterile centrifuge tubes containing 30 mL of sterile water for the isolation of the surface actinomycete” change to Preparation of samples, the insects were placed into sterile centrifuge tubes containing 30 mL of sterile water for the isolation of the surface actinomycetes (Chevrette et al. 2019).
Lines 114-115; “Based on morphological characteristics and molecular identification, 60 selected strains were screened for small-scale fermentation to isolate actinomycetes with antibacterial activity”, the morphological data could not use to select the interesting actinomycetes for antibacterial activity analysis, but should be selected all isolates to ferment the metabolites.
Lines 137-138; “The crude extracts were subjected to silica gel column chromatography, eluted sequentially with varying ratios of dichloromethane and methanol”. Ratio of the solvent system to separated crude?
Lines 139-140; Fractions were monitored by thin-layer chromatography (TLC), and further purification was achieved using Sephadex LH-20 column chromatography and high performance liquid chromatography (HPLC) to obtain purified compounds.
- Explain the steps to obtain each fraction, what are the mobile systems were used with TLC, Sephadex LH-20 and HPLC? (Add the data in supplementary information). Number and yield (g) of the isolated compounds?

Validity of the findings

Foe basic characteristic, should be exhibit the morphological data of the isolates such as color of mycelium, spore formation and cultural data on the media.
Lines 192-195; Apart from the dominant strain S. pratensis, which accounted for 30% of gut isolates, the distribution of other strains was relatively even. In contrast, among the 21 strains isolated from the cuticle, only 4 species were identified, with S. cavourensis (43%) and S. pratensis (48%) being overwhelmingly dominant.
- In the results of species level, that base on 16S rRNA gene analysis the strain could not identify to the species (S. pratensis, S. cavourensis), but it showed only closely relate species by using the similarity values. In addition, the results of molecular identification should be confirmed by phylogenetic tree to determine the cluster group of the strains that related with the similarity values of 16S rRNA gene.

Additional comments

Supplementary Data
Figure S1. Colony morphology of part symbiotic actinomycetes.
- Add isolate code on the figure.
-
Table S2. “Phylogenetic analysis of cultivable actinomycetes associated with P. fuliginosa” change to 16S rRNA similarity values of cultivable actinomycetes isolates with closely related species. And 16S rRNA gene accession no. of actinomycete isolates (86 isolates), and should be deleted the column of completeness data (%) from the Table.

Reviewer 5 ·

Basic reporting

The manuscript "Diversity, antibacterial and phytotoxic activities of actinomycetes associated with Periplaneta fuliginosa" by Q. Liu et al. is a research paper devoted to the study of culturable actinobacteria isolated from cuticle and gut of cockroach Periplaneta fuliginosa. The text is written clear and easy to read, well illustrated. English language fine, no critical issues detected.

Minor corrections needed:
line 40: polyketides, macrolides, quinolones, - the list seems strange: macrolides is a subtype of polyketides; quinolones are not a biosynthetic item, it's about chemical structure;
line 55: application(Seabrooks - add space;

The references is partially non-actual. For example, I could recommend two fresh reviews actual for the current work:
Grundmann C.O., Guzman J., Vilcinskas A., Pupo M.T. The insect microbiome is a vast source of bioactive small molecules. Nat. Prod. Rep., 2024, 41, 935-967. https://doi.org/10.1039/D3NP00054K
Olano C., Rodríguez M. Actinomycetes Associated with Arthropods as a Source of New Bioactive Compounds. Curr. Issues Mol. Biol. 2024, 46, 5, 3822-3838. https://doi.org/10.3390/cimb46050238

Experimental design

The sample collection, processing and isolation of axenic cultures were described by authors. Isolated cultures were tested by antibacterial and plant growth inhibition activity. One isolate, close to Nocardiopsis alba by 16S rRNA gene sequence, was studied more detailed and several active compounds were isolated and characterized from the culture extract.
1. The authors use Nocardiopsis alba ZLC-87 designation for the described isolate but this based only on the analysis of the 16S rRNA gene sequence. That's not enough for taxonomic assignement. So, the authors should provide additional data - multi-locus sequence analysis of housekeeping (essential) genes, at least, or polyphasic taxonomic analysis (biochemical, morphological etc.).
2. Experimantal data on antimicrobial activity should be corrected. The data like "zone of inhibition (ZOI) of 14.67 mm" is absolutely incorrect. It's based on the method's accuracy: the variation of the ZOI in one experiment could reach a few mm due to complex character of diffusion and other factors. The precision like "0.01 mm" is not available. Please, delete the values from the abstract at all, and correct the values presented in the other sections. Please, add the information about used dosage (mkg/disc).

Validity of the findings

Conclusions are well supported by the presented data, I have not any questions about.

Additional comments

The work could be accepted for publication after serious revision.

---

## Round 0.2 · Major Revisions

Please address the remaining concerns of the reviewer and amend manuscript accordingly.

Reviewer 5 ·

Basic reporting

All recommendations were implemented into revised version of the manuscript.

Experimental design

1. The authors are familiar with standard methods of actinobacterial taxonomy, but it has been ignored. The designation Nocardiopsis alba ZLC-87 for the described isolate is incorrect due to it's based only on the analysis of the 16S rRNA gene sequence. So, there is 2 ways to avoid the problems:
- Use "Nocardiopsis sp. ZLC-87" designation for the isolate;
- Provide additional data for taxonomic description of the isolate: morphological, biochemical and/or genetical. For examples you could see https://www.dsmz.de/microorganisms/wink_pdf/Actinomethods.pdf

2. Please, add the information about used dosage (ug/disc) in the abstract.

Validity of the findings

Conclusions are well supported by the presented data, I have not any questions about.

Additional comments

The work could be accepted for publication after revision.

---

## Round 0.3 · accepted · Accept

All remaining concerns of the reviewer were addressed and the revised version is acceptable now.

Reviewer 5 ·

Basic reporting

The manuscript "Diversity, antibacterial and phytotoxic activities of actinomycetes associated with Periplaneta fuliginosa" by Q. Liu et al. is a research paper devoted to the study of culturable actinobacteria isolated from cuticle and gut of cockroach Periplaneta fuliginosa. The text is written clear and easy to read, well illustrated. English language fine, no critical issues detected. The text was improved by authors according recommendations.

Experimental design

The sample collection, processing and isolation of axenic cultures were described by authors. Isolated cultures were tested by antibacterial and plant growth inhibition activity. One isolate, close to Nocardiopsis alba by 16S rRNA gene sequence, was studied more detailed and several active compounds were isolated and characterized from the culture extract.
I guess it will be more adequate to add microscopic observations of the studied Nocardiopsis isolate, but in the current form the manuscript contains enough information for publication. The work could be accepted for publication in the current form.

Validity of the findings

Conclusions are well supported by the presented data.